# Effect of Polymer Ether Ketone Fibers on the Tribological Properties of Resin-Based Friction Materials

**DOI:** 10.3390/ma16052094

**Published:** 2023-03-03

**Authors:** Lekai Li, Zichao Ma, Guoqin Liu, Jin Tong, Wei Song, Lili Ren, Tianjian Tong, Yunhai Ma

**Affiliations:** 1Key Laboratory of Bionic Engineering (Ministry of Education), College of Biological and Agricultural Engineering, Jilin University, Changchun 130022, China; 2Weihai Institute for Bionics, Jilin University, Weihai 264200, China; 3Department of Mechanical Engineering, The Pennsylvania State University, Philadelphia, PA 16802, USA; 4Department of Agricultural and BioSystem Engineering, Iowa State University, Ames, IA 50010, USA

**Keywords:** polymer ether ketone fibers, friction materials, intelligent, secondary plateaus

## Abstract

Resin-based friction materials (RBFM) are widely used in the fields of automobiles, agriculture machinery and engineering machinery, and they are vital for safe and stable operation. In this paper, polymer ether ketone (PEEK) fibers were added to RBFM to enhance its tribological properties. Specimens were fabricated by wet granulation and hot-pressing. The relationship between intelligent reinforcement PEEK fibers and tribological behaviors was investigated by a JF150F-II constant-speed tester according to GB/T 5763-2008, and the worn surface morphology was observed using an EVO-18 scanning electron microscope. The results showed that PEEK fibers can efficiently enhance the tribological properties of RBFM. A specimen with 6 ωt% PEEK fibers obtained the optimal tribological performance, the fade ratio was −6.2%, which was much higher than that of the specimen without the addition of PEEK fibers, the recovery ratio was 108.59% and the wear rate was the lowest, which was 1.497 × 10^−7^ cm^3^/(Nm)^−1^. The reason for the enhancing tribological performance was that, on the one hand, PEEK fibers have a high strength and modulus which can enhance the specimens at lower temperatures; on the other hand, molten PEEK at high temperatures can also promote the formation of secondary plateaus, which are beneficial for friction. The results in this paper can lay a foundation for future studies on intelligent RBFM.

## 1. Introduction

Phenolic resins are widely used in the industries of adhesives, flame retardant materials and friction materials because of their excellent acid resistance, mechanical properties and high temperature resistance [1,2]. However, the poor wear resistance and low impact resistance have severely limited their applications [3,4]. A great deal of studies have shown that the comprehensive properties of resin-based friction materials (RBFM) can be improved with reinforced fibers [5,6]. Bamboo fibers [7,8], mineral fibers [6,9], carbon fibers [10,11,12] and corn stalk fibers [13,14] have successfully improved their tribological properties. In particular, polymer ether ketone (PEEK) fibers provide a promising solution for the realization of intelligent RBFM, with their excellent mechanical properties, tribological properties and temperature sensitivity [15].

PEEK is a kind of semi-crystalline polymer that has an excellent temperature resistance (the glass transition temperature is 143 °C and the melting point is 343 °C) [16]. Its thermodynamic properties can be used as temperature perception components in RBFM at high temperatures (350 °C) and then regulate the friction interface microstructure during friction. Due to the high temperature resistance, molten PEEK can cover on hard particles to prevent them from damaging samples. It can also effectively adhere wear debris and promote the formation of secondary plateaus, thus improving the tribological properties. PEEK has high mechanical properties [17] and self-lubricating properties [18] and an outstanding fatigue resistance to alternating stresses [19]. Thus, PEEK has been used as a popular reinforcement to enhance the mechanical and tribological properties of RBFM.

PEEK has good tribological properties. As a reinforcement, Crosslinking Solidification between PEEK and the phenolic matrix has a negative influence for tribological properties and temperature perception, thus limiting its further application [20]. In addition, PEEK brings more complex interfaces between PEEK and other components, and it will reduce the strength of samples. In recent years, structure design has been a vital method for solving this problem. For example, Yucheng Liu [21] fabricated RBFM by wet granulation; the friction coefficient (COF) and wear rate were 17.85–23.9% higher and 8.62–61.27% lower than those obtained from powders, respectively. In fact, although granulation does not change the composition of a specimen, it modifies the physical structure. The enhancement of tribological properties could be attributed to the fact that wet granulation forms a hard shell, consisting of a high-density water/powder layer on the particles’ surface. Yanwen Yang [22] et al. prepared polytetrafluoroethylene @ phenolic resin composites with a core/shell structure, which successfully solved the problem of the poor tribological performance resulting from polytetrafluoroethylene adhesion. The wear rate has been reduced to one-fifteenth of that of the original. The improvement in wear resistance is mainly due to the formation of uniform and integrate transfer film. Currently, the structure design of friction materials is a popular method for improving tribological performance, and it can also produce unexpected synergistic effects in enhancing tribological properties [13,20,23]. Therefore, the structural design of RBFM is an effective strategy for solving the problem of crosslinking solidification and enhancing its tribological properties.

An effective method for the structure design of composites is the granulation technique [24]. The components addition order during granulation and the particle size can effectively control the internal structure of RBFM [13]. Xiaoyang Wang et al. [24] granulized carbon black and then added it to copper–iron friction materials to enhance the wear resistance. The reduction in the wear rate was mainly attributed to the uniform dispersion of carbon black. Yucheng Liu et al. [13] prepared corn stalk fiber-reinforced friction materials by wet granulation. The improvement in fade resistance was attributed to the fact that wet granulation can effectively reduce the resin content in friction materials, thus alleviating matrix softening and thermal decomposition at high temperatures. Lian et al. [25] prepared Cu/graphene oxide (GO)-Ti_3_AlC_2_/Cu composites with a sandwich structure. During friction, GO and Ti_3_AlC_2_ synergistically promote the formation of a continuous, compact and lubricating tribo-layer on the worn surface and enhance the wear resistance. Lekai Li [20] investigated the tribological properties of PEEK powders reinforcement RBFM; the results showed that PEEK powders can enhance the high temperature resistance of RBFM, but it does not have a positive influence on the tribological performance at low temperatures. Even though there have been many studies on strengthening RBFM, there have been no reports about intelligent reinforcement PEEK fibers, which can perceive temperature and regulate the tribological behaviors of RBFM.

This paper presents an intelligent friction material that can regulate the microstructure of the friction interface through the perception of temperature. The specimens are fabricated by step feeding, wet granulation and particle coating technology and hot-pressing, which can physically isolate the phenolic resin from PEEK fibers to prevent crosslinking solidification. Specimens were subjected to tribological tests and worn surface characterization to study the relationship between the tribological behavior and the PEEK fiber content, which could provide data information for product development in industry and lay a foundation for the development of intelligent tribological materials.

## 2. Materials and Methods

### 2.1. Raw Materials

The content of RBFM in this paper is shown in Table 1, and the information of raw materials is shown in Table 2.

### 2.2. Fabrication of Specimens

PEEK fibers were treated with Silane Coupling Agent (SCA, Jinan Xingfeilong Chemical Co. LTD). The composition of the SCA solution is shown in Table 3. The PEEK fibers were treated in the SCA solution for 60 min at 25 °C and then dried for 12 h at 90 °C in a Heat-Treated Case (JF980S, Wangda, Changchun, China).

Figure 1 presents the fabrication process of the specimen.

The first step was the mixing of raw materials. Fibers including PEEK fibers, Sepiolite fibers and Compound Mineral fibers were thrown into an Electrical Blender (JF801S, Wangda, Changchun, China) for 3–5 min to increase dispersion. After dispersion, all the other compositions except for phenolic resin were thrown into a Compact Rake Blender (JF810, Wangda, Changchun, China) for 8–10 min to obtain the mixture of the raw materials.

The second step was wet granulation, which can separate PEEK fibers and phenolic resin physically to avoid crosslinking solidification. Figure 2 shows the granulation process in this study. The total quantity of Absolute Ethyl Alcohol was about 40 ωt% of the mixture. The granulation device was a Laboratory Tumbling Granulator (JF805R, Wangda, Changchun, China), and the drying device was a Heat-Treated Case (JF980S, Wangda, Changchun, China).

The third step was hot-pressing. Granules were molded for 10 min at 160 °C under 45 MPa by a hot compression machine (JFY50, Wangda, Changchun, China), according to our previous study [13,20,21]. To release volatiles, three intermittent ‘breathings’ were carried out during hot-pressing. To remove the remaining stress after hot-pressing, the samples were heat-treated by a Heat-Treated Case (JF980S, Wangda, Changchun, China), and the temperature is shown in Figure 3.

### 2.3. Testing Methods and Equipment

The tribological performance of RBFM was tested using a Constant-Speed Tester (JF150F-II, Wangda, China) according to GB/T 5763-2008. Figure 4 presented the schematic of the JF150F-II Constant-Speed Tester. A friction disc was driven by an electric motor at a constant speed of 480 rpm, whose hardness was from HB 180 to HB 220. The temperature of the rotating disc was regulated mainly by cooling water and a thermoelectric couple. The samples were pressed by a constant normal pressure of 0.98 MPa, and the samples would be polished with abrasive papers before each test. The COF and wear rate would be tested at 100 °C, 150 °C, 200 °C, 250 °C, 300 °C and 350 °C during the fade test and would be tested at 300 °C, 250 °C, 200 °C, 150 °C and 100 °C during the recovery test. Each specimen would be conducted for five repetitive tests.

The COF *μ* of the specimens was calculated according to Equation (1) [13], and the wear rate ΔV was defined using Equation (2) [13].
(1)$$\mu =\frac{f}{F_{N}}$$
(2)$$\Delta V=\frac{1}{2*\pi *r}*\frac{A}{n}*\frac{d_{1}-d_{2}}{f}$$
where *f* represents the friction force (N); *F_N_* is the normal pressure (N); *r* is the distance between the rotation center and the sample center; *n* is the number of revolutions (5000); *A* is the surface area (*A* = 625 mm^2^); *d*_1_ and *d*_2_ are the initial and final thickness of the sample, respectively.

The tribological test was divided into two stages: a fade test and recovery test. The fade ratio (*F_Fade_*) and recovery ratio (*F_Recovery_*) are calculated according to Equation (3) and Equation (4), respectively.
(3)$$F_{Fade}=\frac{\mu_{F100^{{}^{\circ}}C}-\mu_{F350^{{}^{\circ}}C}}{\mu_{F100^{{}^{\circ}}C}}\times 100\%$$
(4)$$F_{Recovery}=\frac{\mu_{R100^{{}^{\circ}}C}}{\mu_{F100^{{}^{\circ}}C}}\times 100\%$$
where *μ*_F100°*C*_ and *μ*_F350°*C*_ represent the COF in the fade test at 100 °C and 350 °C, respectively, and *μ*_*R*100°*C*_ represents the COF in the recovery test at 100 °C.

After the friction test, the worn surface morphology was observed by an EVO-18 Scanning Electron Microscope (SEM; ZEISS, Jena, Germany) at 20 kV.

## 3. Result and Discussion

### 3.1. Microstructure of PEEK Fibers

The microscopic morphology of the PEEK fiber is shown in Figure 5. It has been reported in the literature that the SCA between organics and inorganics plays the role of a bridge, which can strengthen the interface [27]. From the result in Figure 5, it can be inferred that the SCA can change the morphology of PEEK fibers. The surface of PEEK fibers that are not SCA-treated (Figure 5a) is relatively smooth, with slight folds. Fibers with a smooth surface had a difficult time forming a good interface with other components; they would be pulled out easily under shear force during friction, thus causing abrasion [28,29]. However, PEEK fibers that were SCA-treated presented a rougher surface (Figure 5b). On the one hand, the rough PEEK fibers can form a mechanical interlock with other components [28,30]. On the other hand, SCA formed a strong interface between PEEK fibers and other components, which can also enhance the tribological properties [31].

### 3.2. Fade Resistance

The result of the fade test was summarized in Figure 6. In general, the addition of PEEK fibers reduces the COF at low temperatures, and it also reduces the COF sensitivity to temperature. The result can be attributed to PEEK fibers’ relatively high physical characteristics, such as the strength, the modulus and the immobilized interfacial zone around the fibers, which dissipates most of the braking stress during braking, thus resulting in a lower COF [12,32].

From the results in Figure 6a, it can be seen that the COF of RBFM was influenced by the number of PEEK fibers. As the temperature increased, the COF increased initially and then decreased with the addition of a small number of PEEK fibers (2 ωt%, 4 ωt%). When a larger number of PEEK fibers was added (6 ωt%, 8 ωt%), the COF decreased initially, then increased and finally decreased with the rise in temperature. The decrease in the COF at high temperatures was attributed to phenolic matrix softening and thermal decomposition [28,33]. The PEEK fibers had a different influence with the PEEK powders on COF [20]. The addition of PEEK fibers can further improve the tribological properties of the composites because of the high strength and modulus of fibers compared with those of powders.

From the result in Figure 6b, it can be inferred that PEEK fibers can reduce the COF sensitivity to temperature, and they can also improve the COF stability by 32.9–63.8%. Specifically, RBFM-2 has the lowest COF sensitivity to temperature. Compared to other fibers, PEEK fibers provide a more stable COF [11,34]. On the one hand, this is partly due to the high strength and modulus of PEEK fibers, which maintain the composite with a high strength; on the other hand, the adhesion of molten PEEK fibers in the friction interface helps to form secondary plateaus, which provide a stable friction.

Combining the results in Figure 6b and Figure 7, it can be summarized that PEEK fibers can effectively enhance the fade resistance of RBFM. RBFM-1 maintained a more stable COF at lower temperatures (100 °C). As the braking continues, no significant decrease in the COF occurred (Figure 7a); however, at higher temperature, the softening and thermal decomposition of the phenolic matrix resulted in a decrease in strength. Thus, the COF showed a continuous decrease [28,33]. With the addition of a smaller number of PEEK fibers (2 ωt%, 4 ωt%), the COF faded more slightly (as shown in Figure 7b,c); it was 6.5% and 7.0% lower than that of RBFM-1. A smaller number of PEEK fibers meant less molten PEEK on the friction interface during friction, which could not effectively adhere wear debris to form secondary plateaus, thus causing a heat fade. With the increase in the PEEK fibers content (6 ωt%, 8 ωt%), more molten PEEK on the friction interface can adhere wear debris efficiently, which promoted the formation of secondary plateaus and provided more continuous and stable friction. As a result, the COF remained in a stable state (Figure 7d,e). This trend is in agreement with our earlier-published study [35].

### 3.3. Recovery Property

Figure 8 presented the result of the recovery test. It can be inferred from Figure 8a that PEEK fibers could also effectively influence the COF in the recovery test. As the temperature decreases, the trend of the COF varies with different numbers of of PEEK fibers. Specifically, RBFM-1 showed a gradually increasing trend of the COF. The lower COF at high temperatures is caused by the reduction in strength resulting from the softening and thermal decomposition of the phenolic matrix [36,37]. With the addition of PEEK fibers, molten PEEK fibers on the friction interface help to adhere wear debris and promote the formation of continuous secondary plateaus under normal pressure; the high modulus of PEEK fibers can also reduce the heat fade. Therefore, the COF of specimens with PEEK fibers changes relatively smoothly as the temperature decreases [20,38,39].

Combining the results in Figure 8b and Figure 9a, it can be summarized that, during the recovery test, RBFM-1 still had the highest COF fluctuation, which was caused by the low resistance to shear stress [26]. With the addition of PEEK fibers, the COF sensitivity increased initially, then decreased and finally increased; the COF sensitivity of RBFM-2 was the lowest, and it was 67.36% lower than that of RBFM-1. On the one hand, the high temperature (350 °C) during the fade test modified the interface between PEEK fibers and other components and enhanced the interface strength [40]. On the other hand, since PEEK has a low melting point (343 °C), the addition of excessive PEEK fibers tends to cause an increase in the friction interface softening, which, in turn, enhances the COF sensitivity to temperature [41].

As shown in Figure 8b, during the recovery test, the PEEK fibers reduced the recovery performance to a certain extent. RBFM-1 had the best recovery performance of 124.1%. After the fade test, it was difficult for wear debris to be compacted and form secondary plateaus; there is more wear debris on the friction interface, which would hinder the relative slide between the counterpart and the specimen, thus causing a higher COF (Figure 9a) [42,43]. For the other samples, the wear debris on the friction interface would be compacted as the secondary plateaus, thus causing little wear debris on the friction interface such that the COF was low, resulting in a lower recovery rate (Figure 9b,e) [44,45,46].

### 3.4. Wear Resistance

The result of the wear rate is shown in Figure 10. As is evident, the disc temperature can significantly affect the wear rate of the samples (Figure 10a). The high wear rate was related to the phenolic matrix softening and the thermal decomposition at high temperatures [36,43]. This is consistent with our previous research [26,34]. From Figure 10a, it can be seen that the wear rate increases with increasing temperatures. Between 100 and 250 °C, the wear rate increases slowly; however, it increases more rapidly from 250 °C to 350 °C. The phenolic matrix mainly undergoes matrix softening at low temperatures so that the specimens can still maintain a high strength [13]. However, the phenolic matrix would decompose at high temperatures, which would cause a rapid decrease in strength; thus, the wear rate increases rapidly [36].

Figure 10a shows that PEEK fibers can reduce the wear rate at lower temperatures (100–200 °C). PEEK fibers play a load-bearing role, which can enhance the strength of the composite at low temperatures, increasing the wear resistance. As the temperature increased (250–300 °C), the specimens with a smaller number of PEEK fibers (2 ωt%, 4 ωt%) had a higher wear rate than RBFM-1, and the samples with more PEEK fibers (6 ωt%, 8 ωt%) had a lower wear rate. One possible explanation is that the strength of PEEK fibers decreases sharply when the temperature is much higher than the Glass Transition Temperature, resulting in a lower load-bearing capacity. At 350 °C, the PEEK fibers at the friction interface melted and adhered wear debris, promoting the formation of continuous and dense secondary plateaus, which was important for reducing the wear rate [21,45].

Figure 10b reflects the total wear rate of the samples. PEEK fibers can strengthen the specimens at low temperatures and promote the formation of secondary plateaus at high temperatures, which can enhance the wear resistance [20]. Thus, the wear rate decreased as the PEEK fibers content increased from 0 ωt% to 6 ωt% (Figure 10b). However, too many PEEK fibers meant a more complicated interface with other components, which would reduce the strength of the samples, resulting in an increase in the wear rate (from 6 ωt% to 8 ωt%; Figure 10b). It can also be inferred that PEEK fibers can reduce the wear rate from 24.7% to 51.1%, and RBFM-4 obtained the lowest wear rate of 1.497 × 10^−7^ cm^3^ × (N × m)^−1^, which was 51.1% lower than that of RBFM-1.

In summary, the addition of 6 ωt% PEEK fibers provides the best compromise of tribological properties. On one hand, the addition of 6 ωt% PEEK fibers leads to a higher fade resistance and recovery performance, which can ensure the stability of the braking; on the other hand, it can also significantly improve the service life of RBFM and ensure the safety of braking.

### 3.5. Morphology of the Worn Surface

Friction is determined by the contact area condition, which is composed of hard materials and the compaction of wear debris around them (called primary and secondary plateaus, respectively) [38,47]. Plateaus, microcracks, abrasive debris and deformation on the worn surface provide useful information for tribology analysis. As Zhang and Sun proposed, a micro-morphological analysis of the worn surface was performed, which can help in understanding the wear mechanism in order to investigate the relationship between tribological behaviors and PEEK fibers [48,49].

The microscopic morphology of the worn surface of RBFM-1 was shown in Figure 11a. The worn surface of RBFM-1 appears to be very rough, with a large number of bare fibers and large flake pits presented and a smaller number of primary and secondary plateaus. RBFM-1 appeared to have severe adhesive wear [1], which was similar with our previous study [20]. On one hand, the samples without the reinforcement of PEEK fibers have a lower strength and are prone to being damaged during friction [50,51,52]; on the other hand, at high temperatures, molten PEEK fibers can adhere wear debris on the worn surface, which is compacted as secondary plateaus under normal pressure, thus enhancing the wear resistance [20,53].

Figure 12 presents the formation mechanism of the secondary plateaus of PEEK fibers-reinforced friction materials. During friction, a series of physical and chemical changes occur on the worn surface, and parts of materials are detached from the worn surface and form wear debris. Wear debris has a certain obstructive effect on the relative slide between the counterpart and the specimen, and it may also damage the specimen or counterpart [54,55]. During friction, due to the excitation of high temperature, PEEK fibers melt and then encapsulate wear debris on the friction interface, which can prevent hard particles from damaging the friction surface and can also adhere wear debris, promoting the formation of secondary plateaus, forming the protection for friction materials and reducing abrasion [56,57].

The microscopic morphology of the worn surface of RBFM-2 is shown in Figure 11b. When a smaller number of PEEK fibers was added (2 ωt%), the abrasion was lower than that of RBFM-1 because the molten PEEK is not enough to effectively adhere wear debris to form secondary plateaus, and the worn surface is still rougher overall. Bare fibers, more pits and cracks occurred on the worn surface. The wear mechanism is fatigue wear and adhesive wear [20,51], with a small amount of plastic deformation. As the PEEK fibers content increased (4 ωt%), larger and denser secondary plateaus were formed on the worn surface, as shown in Figure 11c. The dense and large secondary plateaus will carry more of a load to protect the specimen [47,58], thus causing decreased abrasion. With the further increase in the PEEK fibers content (6 ωt%), the molten PEEK fibers could further promote the formation of large secondary plateaus, as shown in Figure 11d, corresponding to a lower wear rate (Figure 10). As the PEEK fibers content reached 8 ωt%, although more molten PEEK fibers tended to adsorb more wear debris, the lower strength of the PEEK at high temperatures resulted in secondary plateaus with a lower strength, thus forming plastic deformation and spalling pits, as shown in Figure 11e.

## 4. Conclusions

In summary, intelligent RBFM were fabricated by wet granulation, which could perceive the friction interface temperature and regulated the microstructure of the friction interface during braking. Specimens with 6 ωt% PEEK fibers had the best fade resistance and the lowest wear rate, and the stability of the COF was greatly improved. The main reason is that the high strength and modulus of the PEEK fibers enhanced the strength of the RBFM at lower temperatures, and the molten PEEK at high temperatures could effectively adhere wear debris, thus promoting the formation of secondary plateaus, which provided stable and continuous friction. Thus, PEEK fibers are a promoting intelligent reinforcement of RBFM. In the near future, a further study will be conducted to investigate the mechanical properties of PEEK fibers-reinforced RBFM.

## Figures and Tables

**Figure 1 materials-16-02094-f001:**
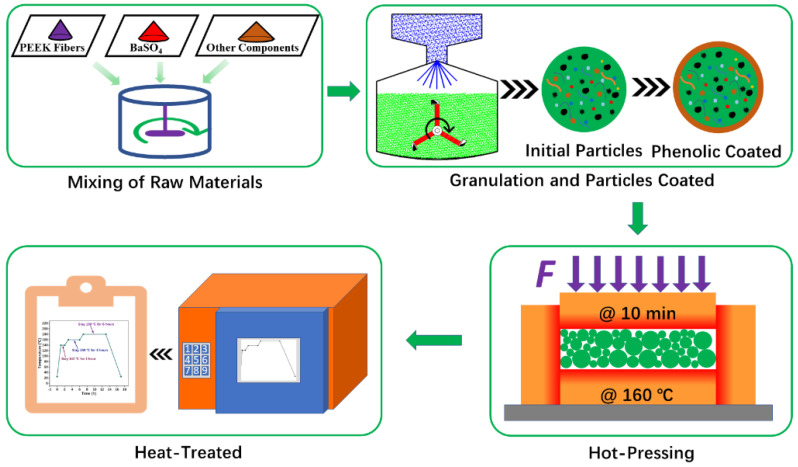
The fabrication process of specimens [13,26].

**Figure 2 materials-16-02094-f002:**
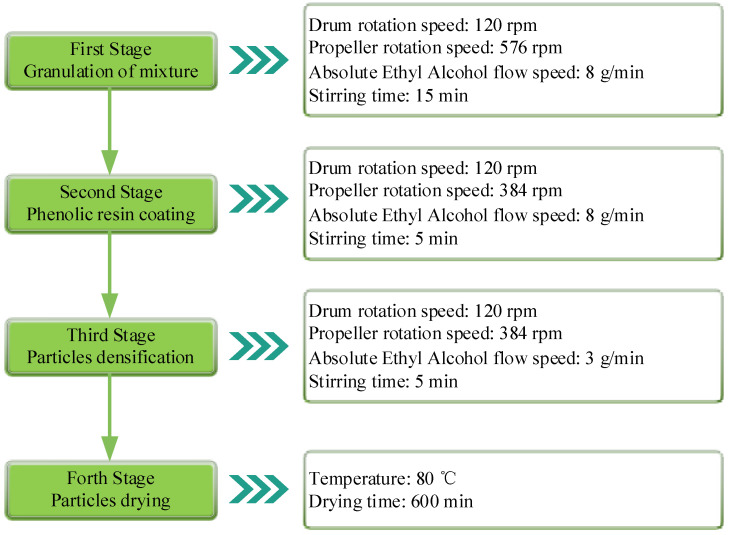
The granulation method of the mixture.

**Figure 3 materials-16-02094-f003:**
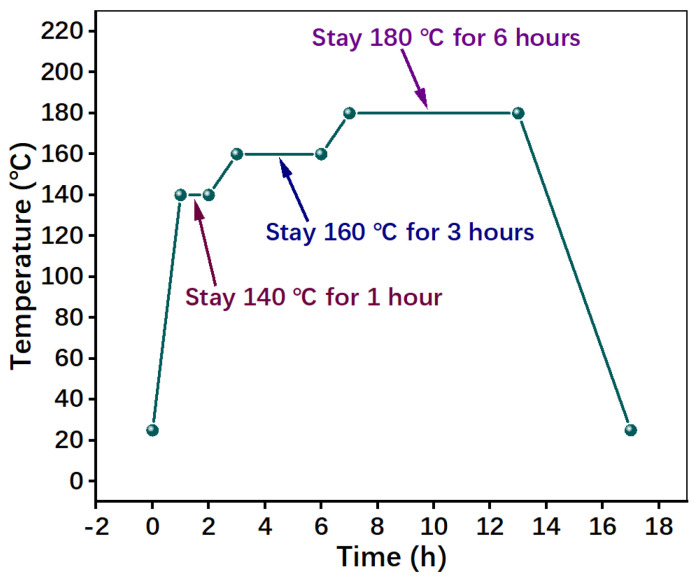
Heat-treatment of specimens.

**Figure 4 materials-16-02094-f004:**
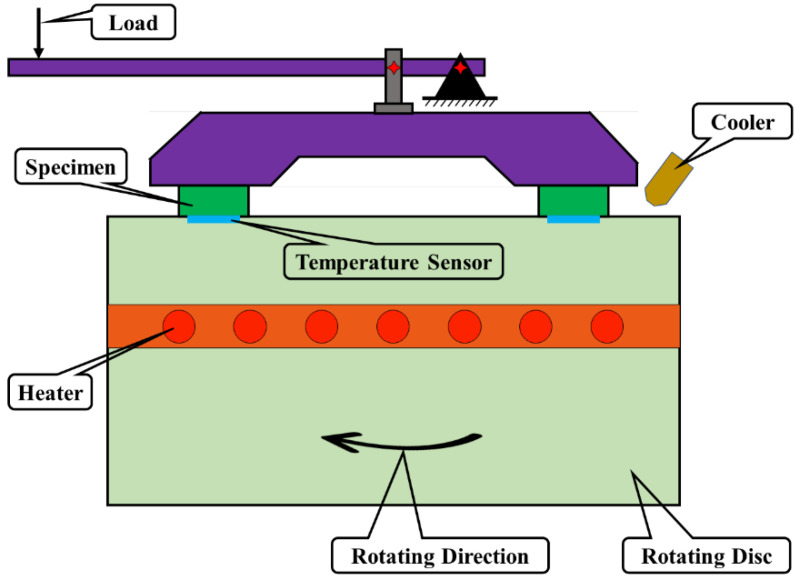
The schematic diagram of the JF150F-II Constant-Speed Tester.

**Figure 5 materials-16-02094-f005:**
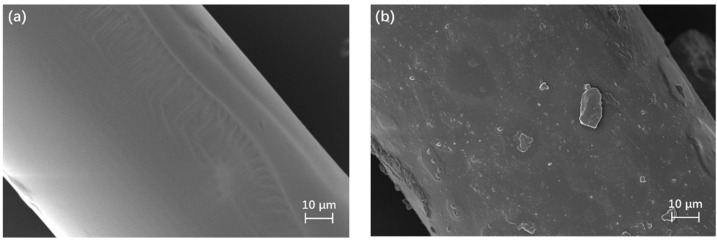
The microscopic morphology of PEEK fibers; (**a**) Unmodified surface of PEEK fibers, (**b**) Surface of PEEK fibers modified with KH550.

**Figure 6 materials-16-02094-f006:**
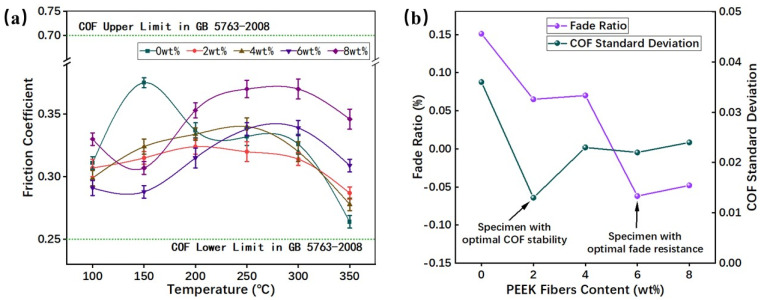
COF of the fade test: (**a**) COF at each temperature, (**b**) Fade ratio and COF Standard Deviation.

**Figure 7 materials-16-02094-f007:**
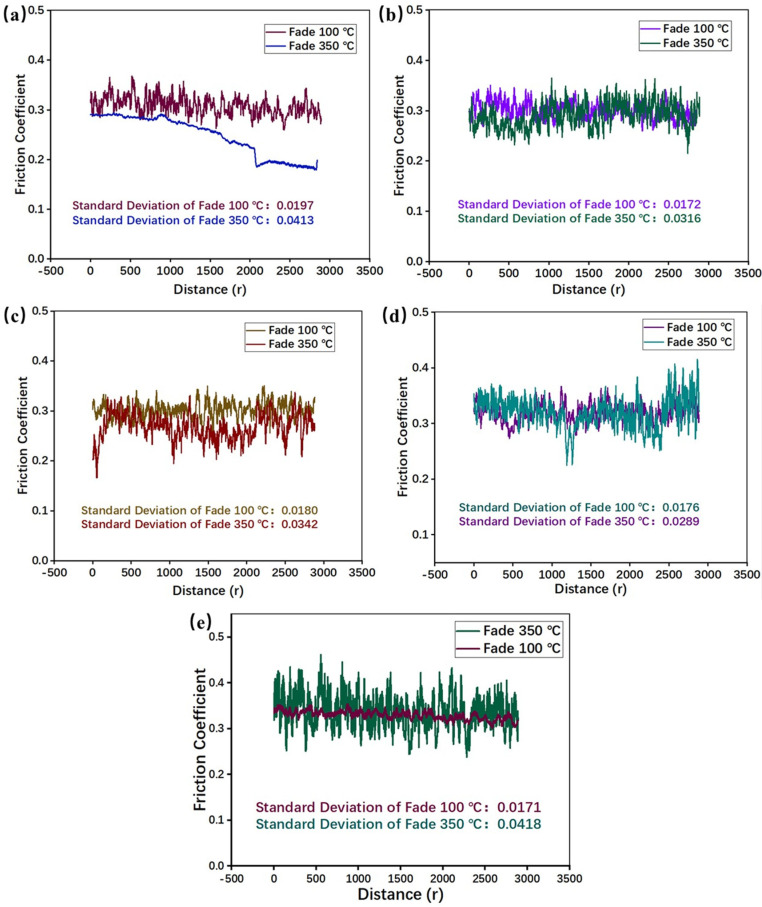
Brake curves of samples during the fade test at 100 °C and 350 °C; (**a**) RBFM-1, (**b**) RBFM-2, (**c**) RBFM-3, (**d**) RBFM-4, (**e**) RBFM-5.

**Figure 8 materials-16-02094-f008:**
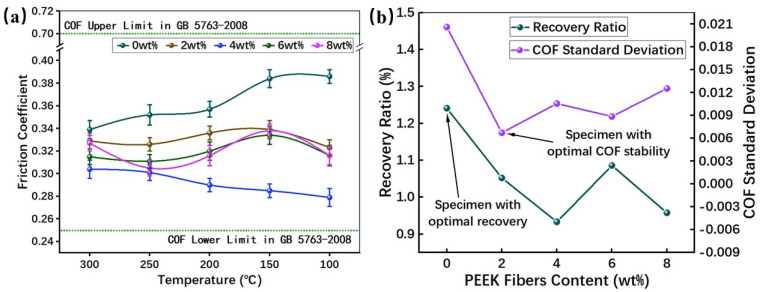
COF of the recovery test; (**a**) COF at each temperature, (**b**) recovery ratio and COF Standard Deviation.

**Figure 9 materials-16-02094-f009:**
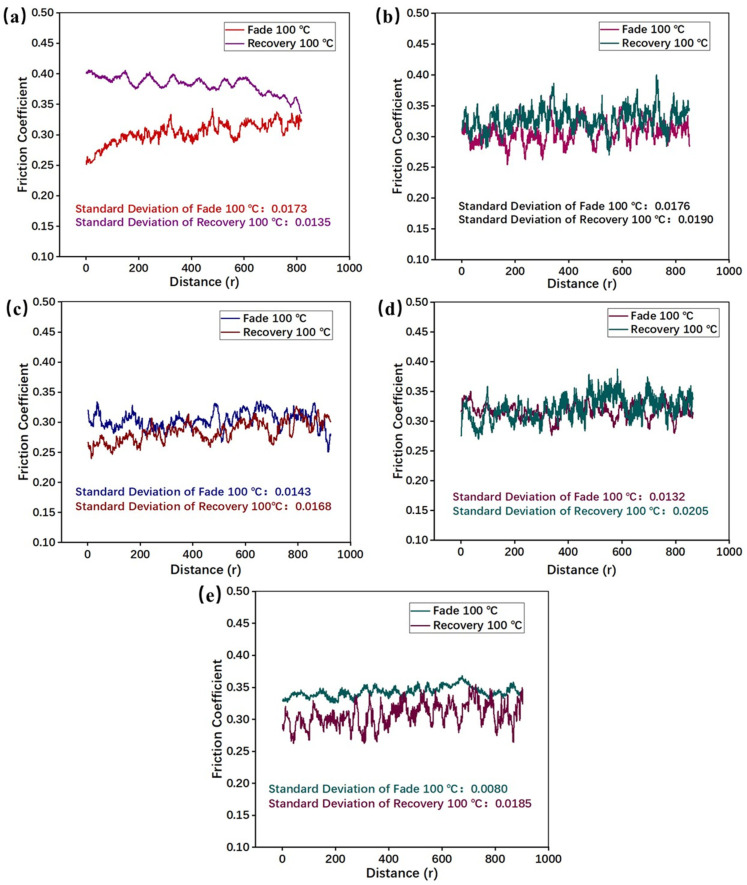
Brake curves of samples during the recovery test at 100 °C and the fade test at 100 °C; (**a**) RBFM-1, (**b**) RBFM-2, (**c**) RBFM-3, (**d**) RBFM-4, (**e**) RBFM-5.

**Figure 10 materials-16-02094-f010:**
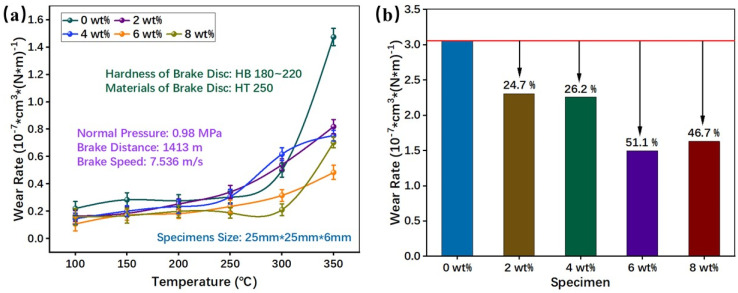
Wear rate of each specimen; (**a**) Wear rate at each temperature, (**b**) Total Wear Rate.

**Figure 11 materials-16-02094-f011:**
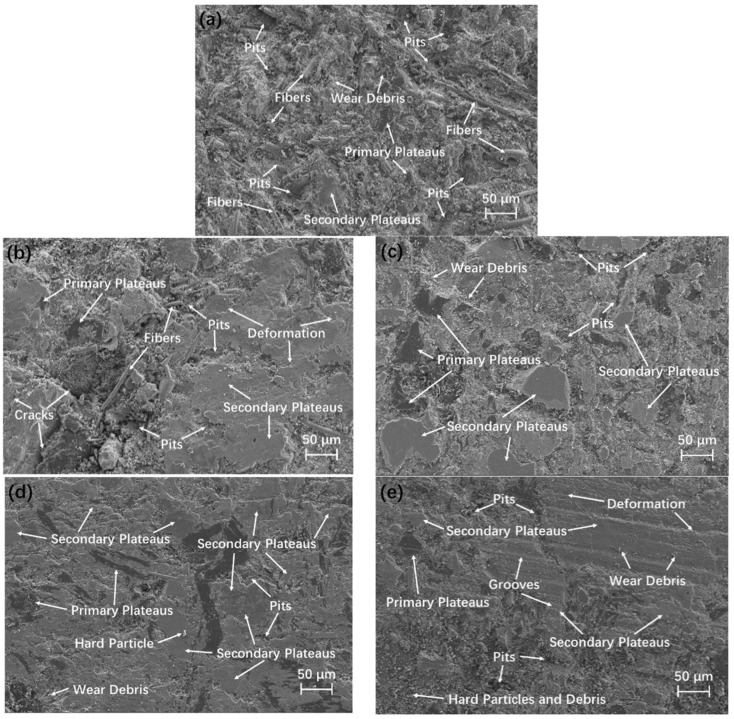
SEM micrographs of the worn surface; (**a**) RBFM-1, (**b**) RBFM-2, (**c**) RBFM-3, (**d**) RBFM-4, (**e**) RBFM-5.

**Figure 12 materials-16-02094-f012:**
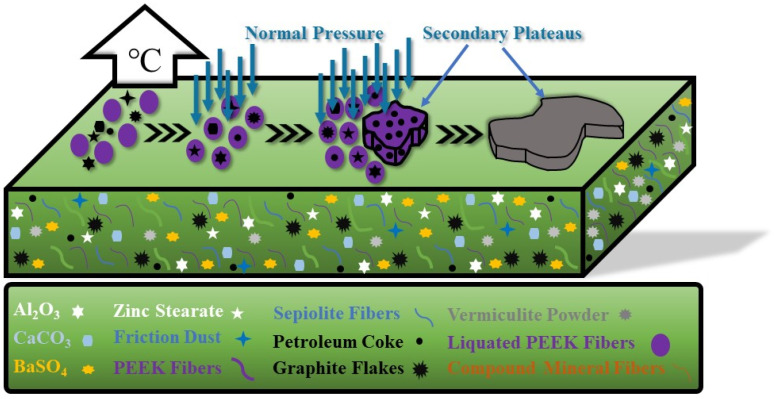
The formation of secondary plateaus.

**Table 1 materials-16-02094-t001:** The content of RBFM of each specimen.

Raw Materials (by ωt%)	Specimens				
	RBFM-1	RBFM-2	RBFM-3	RBFM-4	RBFM-5
PEEK Fibers	0	2.00	4.00	6.00	8.00
Sepiolite Fibers	5.00	5.00	5.00	5.00	5.00
Compound Mineral Fibers	20.00	20.00	20.00	20.00	20.00
Phenolic Powders	9.00	9.00	9.00	9.00	9.00
Graphite	8.00	8.00	8.00	8.00	8.00
Petroleum Coke	7.00	7.00	7.00	7.00	7.00
Aluminum Oxide	6.00	6.00	6.00	6.00	6.00
Friction Dust	2.00	2.00	2.00	2.00	2.00
Calcium Carbonate	13.00	13.00	13.00	13.00	13.00
Vermiculite Powder	5.00	5.00	5.00	5.00	5.00
Barium Sulfate	24.00	22.00	20.00	18.00	16.00
Zinc Stearate	1.00	1.00	1.00	1.00	1.00

**Table 2 materials-16-02094-t002:** The information of raw materials.

Raw Materials	Size (mesh)	Shape	Supply	Properties
PEEK Fibers	Diameter: 0.1 mm; Length: 3 mm	Fiber	Changzhou Chuangying New Material Technology Co., LTD. Changzhou, China	Glass transition temperature: 143 °C; Melting point: 343 °C
Sepiolite Fibers	Diameter: 0.15 mm; Length: 2.5 mm	Fiber	Lingshou Jiasuo Building Materials Processing Co. LTD. Shijiazhuang, China	High temperature resistance
Compound Mineral Fibers	Diameter: 0.2 mm; Length: 3 mm	Fiber	Shijiazhuang Mayue Building Materials Co. LTD. Shijiazhuang, China	Apparent density: 0.13–0.20 g/cm^3^
Phenolic Powders	200 mesh	Irregularity	Henan Borun Casting Material Co. LTD. Zhengzhou, China	Soften temperature: 95–110 °C
Graphite	100 mesh	Flake	Henan Borun Casting Material Co. LTD	Density: 2.1–2.3 g/cm^3^
Petroleum Coke	400 mesh	Irregularity	Shijiazhuang Yuxin Building Materials Co. LTD, Shijiazhuang, China	Density: 1.97–2.15 g/cm^3^
Aluminum Oxide	325 mesh	Globular	Henan Borun Casting Material Co. LTD	Density: 3.9–4.0 g/cm^3^
Friction Dust	100 mesh	Irregularity	Henan Borun Casting Material Co. LTD	Aiming to reduce braking noise
Calcium Carbonate	1250 mesh	Irregularity	Shandong Yusuo Chemical Technology Co. LTD. Linyi, China	Density: 2.8 g/cm^3^
Vermiculite Powder	30 mesh	Irregularity	Lingshou Xuyang Mining Co. LTD. Shijiazhuang, China	Density: 2.5 g/cm^3^
Barium Sulfate	325 mesh	Irregularity	Shandong Yusuo Chemical Technology Co. LTD	Density: 4.3 g/cm^3^
Zinc Stearate	200 mesh	Irregularity	Wuxi Yatai Joint Chemical Co. LTD. Wuxi, China	Density: 1.1 g/cm^3^

**Table 3 materials-16-02094-t003:** The composition of the SCA solution.

Composition	Distilled Water	Absolute Ethyl Alcohol	SCA
Content (by ωt%)	8	72	20

## Data Availability

The datasets generated and/or analyzed during the current study are available from the corresponding author upon reasonable request.

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
