# Peer review of "Effect of Polymer Ether Ketone Fibers on the Tribological Properties of Resin-Based Friction Materials"

_materials, 2023, doi:10.3390/ma16052094_

Round 1

Reviewer 1 Report

I enjoy reading this paper. The contents are good.

My suggestion, the abstract is added with brief information about the background, objectives, research methods, research results and future research continuity.

In the introductory section, the benefits of research for implementation in industry can be added.

At the end of the discussion, the limitations of this research can be explained.

Figure 1 is made clearer and more interesting.

Author Response

My suggestion, the abstract is added with brief information about the background, objectives, research methods, research results and future research continuity.

Reply: Thank you for your constructive suggestion. I have added the brief information about the background, objectives, research methods, research results and future research continuity in the abstract. They were as follows.

Resin based friction materials (RBFM) were widely used in the fields of automobiles, agriculture machinery and engineering machinery, which were vital for the safe and stable operation. In this paper, polymer ether ketone (PEEK) fibers were added to RBFM to enhance its tribological properties. Specimens were fabricated by wet granulation and hot-pressing. Relationship between intelligent reinforcement PEEK fibers and tribological behaviors were investigated by JF150F-II constant-speed tester according to GB/T 5763-2008 and worn surface morphology were observed using an EVO-18 scanning electron microscope. The results showed that PEEK fibers can efficiently enhance the tribological properties of RBFM. Specimen with 6 ωt% PEEK fibers got the optimal tribological performance, the fade ratio was -6.2%, which was much higher than that of specimen without PEEK fibers addition, the recovery ratio was 108.59%, and the wear rate was the lowest which was 1.497×10-7 cm3/(Nm)-1. The reason for the enhancing tribological performance was that on the one hand, PEEK fibers have a high strength and modulus which can enhance the specimens at lower temperature, on the other hand molten PEEK at high temperature can also promote the formation of secondary plateaus which are beneficial for friction. The results in this paper can lay a foundation for future study in intelligent RBFM.

In the introductory section, the benefits of research for implementation in industry can be added.

Reply: Your comments have been accepted. I have added the benefits of research for implementation in industry in the revised manuscript. They were as follows.

This paper presents an intelligent friction material that can regulate microstructure of friction interface through the perception of temperature. The specimens are fabricated by step feeding, wet granulation and particle coating technology and hot-pressing, which can physically isolate phenolic resin from PEEK fibers to prevent crosslinking solidification. Specimens were subjected to tribological tests and worn surface characterization to study the relationship between tribological behavior and PEEK fiber content, which could provide data information for product development in industry and lay a foundation for the development of intelligent tribological materials.

At the end of the discussion, the limitations of this research can be explained.

Reply: Thank you for your constructive suggestions. I have added the limitations of this research at the end of the discussion, they were as follows.

In summary, intelligent RBFM were fabricated by wet granulation, which could percept the friction interface temperature and regulated the microstructure of friction interface during braking. Specimens with 6 ωt% PEEK fibers had the best fade resistance and the lowest wear rate, and the stability of COF was greatly improved. The main reason is that high strength and modulus of PEEK fibers enhanced the strength of RBFM at lower temperature, and molten PEEK at high temperature could effectively adhere wear debris, thus promoting the formation of secondary plateaus, which providing stable and continuous friction. Thus, PEEK fibers are a promoting intelligent reinforcement of RBFM. And in the near future, the further study will be conducted to investigate the mechanical properties of PEEK fibers reinforced RBFM.

Figure 1 is made clearer and more interesting.

Reply: Thank you for your comments. I have made Figure 1 clearer and more interesting in the manuscript.

Reviewer 2 Report

Journal: Materials (ISSN 1996-1944)

Manuscript ID: materials-2246412

The authors presented an article on “Effect of Polymer Ether Ketone Fibers on Tribological Properties of Resin Based Friction Materials”. I think the article is well organized and suitable for the "Materials" journal. But the article will be ready for publication after a major revision. Turnitin similarity rate is 22%. Comments are listed below.

1.      It seems that some references given in the introduction are out of date. More recent references may be offered.

2.      The innovative aspect of the study should be stated in the last paragraph of the introduction. In addition, the difference from similar studies in the literature should be clearly demonstrated.

3.      There is no explanation in the "2.1. Raw materials" section. For example, explanations can be made about materials' shape, size, supply, and properties.

4.      According to which standards were the production parameters of the samples determined?

5.      The wear device of properties and wear parameters are not mentioned in the material and method section. In addition, sample fabrication should be explained in more detail.

6.      Figure labels should be written in the form of "Figure ...(a), (b), (c)" in multiple pictures. For example, Figure 3. The microscopic morphology of PEEK fibers; (a) Unmodified surface of PEEK fibers, (b) Surface of PEEK fibers modified with KH550. All multiple shapes must be corrected.

7.      Is there an error in Figure 7? Temperatures on the figure label are 100 C and 300 C. But in the figures, 100 C and 100 C.

8.      The article contains numerous typographic and language errors. It should be corrected.

9.      The article should be rearranged by taking into account the journal writing rules and citation rules.

*** Authors must consider them properly before submitting the revised manuscript. A point-by-point reply is required when the revised files are submitted.

Author Response

  1. It seems that some references given in the introduction are out of date. More recent references may be offered.

Reply: Thank you for your constructive suggestion. I have offered more recent references in the revised manuscript.

  1. The innovative aspect of the study should be stated in the last paragraph of the introduction. In addition, the difference from similar studies in the literature should be clearly demonstrated.

Reply: Thank you for your comments. Your suggestions have been accepted. I have added the innovative aspect of the study and demonstrated the difference from similar studies in the literature in the revised manuscript at the forth paragraph of Introduction. They were as follows.

Lekai Li[1] investigated tribological properties of PEEK powders reinforcement RBFM, the results showed that PEEK powders can enhance the high temperature resistance of RBFM, but it hasn’t a positive influence on tribological performance at low temperature. Even though there were many studies about strengthening RBFM, there were no reports about intelligent reinforcement PEEK fibers which can percept temperature and regulate tribological behaviors of RBFM.

  1. There is no explanation in the "2.1. Raw materials" section. For example, explanations can be made about materials' shape, size, supply, and properties.

Reply: Your comments have been accepted. I have added the explanation in the "2.1. Raw materials" section in Table 2.

  1. According to which standards were the production parameters of the samples determined?

Reply: Thank you for your constructive suggestions. The samples were fabricated according to our previous studies. I have added it in the revised manuscript. The references are as follows.

[1]   LI L, GAO G, TONG J, et al. Tribological and mechanical behaviours of resin‐based friction materials based on microcrack filling [J]. Biosurface and Biotribology, 2022.

[2]   LI L, ZHUANG J, TONG T, et al. Effect of Wet Granulation on Tribological Behaviors of Cu-Based Friction Materials [J]. Materials, 2023, 16(3).

[3]   LIU Y, WANG L, LIU D, et al. Evaluation of wear resistance of corn stalk fiber reinforced brake friction materials prepared by wet granulation [J]. Wear, 2019, 432-433.

[4]   MA Y, LIU Y, MENON C, et al. Evaluation of Wear Resistance of Friction Materials Prepared by Granulation [J]. ACS Appl Mater Interfaces, 2015, 7(41): 22814-20.

[5]   MA Y, WU S, ZHUANG J, et al. Tribological and physio-mechanical characterization of cow dung fibers reinforced friction composites: An effective utilization of cow dung waste [J]. Tribology International, 2019, 131: 200-11.

  1. The wear device of properties and wear parameters are not mentioned in the material and method section. In addition, sample fabrication should be explained in more detail.

Reply: Thank you for your constructive suggestions. I have added the wear device of properties and wear parameters to the material and method section, and sample fabrication were explained more detailly in the revised manuscript. They were as follows.

Tribological performance of RBFM was tested using Constant-Speed Tester (JF150F-II, Wangda, China) according to GB/T 5763-2008. Figure 4 presented the schematic of JF150F-II Constant-Speed Tester. Friction disc was driven by an electric motor at a constant speed of 480 rpm whose hardness was from HB 180 to HB 220. Temperature of rotating disc was regulated mainly by cooling water and thermoelectric couple. Samples were pressed by a constant normal pressure of 0.98 MPa and samples would be polished with abrasive papers before each test. COF and wear rate would be tested at 100 ℃, 150 ℃, 200 ℃, 250 ℃, 300 ℃ and 350 ℃ during fade test and would be tested at 300 ℃, 250 ℃, 200 ℃, 150 ℃, 100 ℃ during recovery test. Each specimen would be conducted for five repetitive tests.

The first step was the mixing of raw materials.  Fibers including PEEK fibers, Sepiolite fibers and Compound Mineral fibers were thrown into Electrical Blender (JF801S, Wangda, Changchun, China) for 3-5 min to increase dispersion. After dispersion, all the other composition except phenolic resin were thrown into Compact Rake Blender (JF810, Wangda, Changchun, China) for 8–10 min to get the mixture of raw materials.

The second step was wet granulation which can separate PEEK fibers and phenolic resin physically to avoid crosslinking solidification. Figure 2 showed the granulation process in this study. The total quantity of Absolute Ethyl Alcohol was about 40 ωt% of the mixture. The granulation device was Laboratory Tumbling Granulator (JF805R, Wangda, Changchun, China)and the drying device was Heat Treated Case(JF980S, Wangda, Changchun, China).

The third step was hot-pressing. Granules were moulded for 10 min at 160 ℃ under 45 MPa by hot compression machine (JFY50, Wangda, Changchun, China) according to our previous study[1-3]. To release volatiles, three intermittent ‘breathings’ were carried out during hot-pressing. And to remove remaining stress after hot-pressing, samples were heat-treated by Heat Treated Case (JF980S, Wangda, Changchun, China) and the temperature is shown in Figure 3.

  1. Figure labels should be written in the form of "Figure ...(a), (b), (c)" in multiple pictures. For example, Figure 3. The microscopic morphology of PEEK fibers; (a) Unmodified surface of PEEK fibers, (b) Surface of PEEK fibers modified with KH550. All multiple shapes must be corrected.

Reply: Thank you for your constructive suggestion. I have revised the form of figure labels in the manuscript.

  1. Is there an error in Figure 7? Temperatures on the figure label are 100 C and 300 C. But in the figures, 100 C and 100 C.

Reply: Thank you for your comment, I have revised the error in Figure 7 in the manuscript.

  1. The article contains numerous typographic and language errors. It should be corrected.

Reply: Thank you for your constructive suggestions. I have corrected the typographic and language errors in the manuscript.

  1. The article should be rearranged by taking into account the journal writing rules and citation rules.

Reply: Thank you for your suggestions. I have rearranged the manuscript according to the journal writing rules and citation rules.

Reviewer 3 Report

In this paper, the authors present results of reinforcement of resin based friction materials (RBFM) by polymer ether ketone (PEEK) fibers e fabricated by wet granulation method. The results are promising for application of the development technology for practical useful purposes. The paper is well written and follows the standards of technical literature. However, minor modifications are needed to address my questions, as listed herein: 

1. "PEEK has high tribological properties" - what is the meaning of "high" tribological properties?

2. Please rewrite this sentence to make it meaningful - "Crosslinking Solidification with phenolic matrix has severely hindered the perception of temperature and regulation of the friction interface microstructure". 

3. Could you elaborate on possible reasons why the wear rate decreased firstly and then increased?

Author Response

  1. "PEEK has high tribological properties" - what is the meaning of "high" tribological properties?

Reply: Thank you for your comments. The “high” tribological properties means good friction characteristics and wear resistances. Literatures have reported that PEEK composites are widely employed for their good tribological characteristics[1, 2, 3] which is a thermoplastic engineering plastic with excellent mechanical properties and can be used over a large temperature range[1, 4]. The expression of “high tribological properties” was improper and I have changed it as “good tribological properties” in the manuscript.

[1] XIONG D, XIONG L, LIU L. Preparation and tribological properties of polyetheretherketone composites [J]. J Biomed Mater Res B Appl Biomater, 2010, 93(2): 492-6.

[2] THEILER G, GRADT T. Friction and wear of PEEK composites in vacuum environment [J]. Wear, 2010, 269(3-4): 278-84.

[3] Ellinas, K.; Pujari, S. P.; Dragatogiannis, D. A.; Charitidis, C. A.; Tserepi, A.; Zuilhof, H.; Gogolides, E., Plasma micro-nanotextured, scratch, water and hexadecane resistant, superhydrophobic, and superamphiphobic polymeric surfaces with perfluorinated monolayers. ACS Appl Mater Interfaces 2014, 6, (9), 6510-24.

[4] CHANG L, ZHANG Z, YE L, et al. Tribological properties of high temperature resistant polymer composites with fine particles [J]. Tribology International, 2007, 40(7): 1170-8.

  1. Please rewrite this sentence to make it meaningful - "Crosslinking Solidification with phenolic matrix has severely hindered the perception of temperature and regulation of the friction interface microstructure". 

Reply: Thank you for your suggestion. I have rewritten the sentence in the manuscript. It was as follows.

PEEK has good tribological properties. As a reinforcement, Crosslinking Solidification between PEEK and phenolic matrix has a negative influence for tribological properties and temperature perception, thus limited its further application.

  1. Could you elaborate on possible reasons why the wear rate decreased firstly and then increased?

Reply: Thank you for your constructive suggestion. I have elaborate on the reasons why the wear rate decreased firstly and then increased in the manuscript and it was as follows.

Figure 10b reflects the total wear rate of samples. PEEK fibers can strengthen the specimens at low temperature and promote the formation of secondary plateaus at high temperature which can enhance the wear resistance. Thus wear rate decreased with increasing PEEK fibers content from 0 ωt% to 6 ωt% (Figure 10b). However, too much PEEK fibers meant more complicated interface with other components, which would reduce the strength of samples, resulting the increase of wear rate (from 6 ωt% to 8 ωt%, Figure 10b).

Round 2

Reviewer 2 Report

Journal: Materials (ISSN 1996-1944)

Manuscript ID: materials-2246412

Review report R2#

The authors made the desired corrections. Therefore, in my opinion, this article can be accepted for publication in the "Materials" journal in its final form.
